# Antarctic Ozone Depletion between 1960 and 1980 in Observations and Chemistry-Climate Model Simulations

Ulrike Langematz[1], Franziska Schmidt[1], Markus Kunze[1], Gregory E. Bodeker[2], and Peter Braesicke[3]

[1]Freie Universität Berlin, Berlin, Germany
[2]Bodeker Scientific, Alexandra, New Zealand
[3]Karlsruhe Institute of Technology, Karlsruhe, Germany

*Correspondence to*: Ulrike Langematz (ulrike.langematz@met.fu-berlin.de)

**Abstract.** The year 1980 has often been used as a benchmark for the return of Antarctic ozone to conditions assumed to be unaffected by emissions of ozone depleting substances (ODSs), implying that anthropogenic ozone depletion in Antarctica started around 1980. Here, the extent of anthropogenically-driven Antarctic ozone depletion prior to 1980 is examined using output from transient Chemistry-Climate Model (CCM) simulations from 1960 to 2000 with prescribed changes of ozone depleting substance concentrations in conjunction with observations. A regression model is used to attribute CCM modelled and observed changes in Antarctic total column ozone to halogen-driven chemistry prior to 1980. Winter-time Antarctic ozone is strongly affected by dynamical processes that vary in amplitude from year to year and from model to model. However, when the dynamical and chemical impacts on ozone are separated, all models consistently show a long-term, halogen-induced negative trend in Antarctic ozone from 1960 to 1980. The anthropogenically-driven ozone loss from 1960 to 1980 ranges between $26.4 \pm 3.4$ % and $49.8 \pm 6.2$ % of the total anthropogenic ozone depletion from 1960 to 2000. An even stronger ozone decline of $56.4 \pm 6.8$ % was estimated from ozone observations. This analysis of the observations and simulations from 17 CCMs clarifies that while the return of Antarctic ozone to 1980 values remains a valid milestone, achieving that milestone is not indicative of full recovery of the Antarctic ozone layer from the effects of ODSs.

## 1 Introduction

Over the past few decades, the evolution of Antarctic stratospheric ozone has been dominated by chemical depletion due to anthropogenic sources of active chlorine (Cl) and bromine (Br) (e.g., WMO [1990], and references therein). This secular ozone change has been modulated by ozone variations on inter-annual timescales caused by dynamically induced temperature fluctuations (e.g., Huck et al. [2005], Newman et al. [2006]). To attribute changes in stratospheric ozone to depletion by halogens, equivalent effective stratospheric chlorine (EESC) is used as an indicator of the chemical effects of ozone depleting substances (ODSs) (Daniel et al. [1995]). EESC is derived from measurements of ground-based halocarbon concentrations, taking into account their transport times into the stratosphere and their conversion into reactive chlorine ($Cl_y$) and bromine ($Br_y$) (e.g., Solomon and Wuebbles [1995]; Montzka and Fraser [2003]; Newman et al. [2006]).

As a result of the Montreal Protocol on Substances that Deplete the Ozone Layer and its subsequent amendments and adjustments that regulated the production and consumption of halocarbons, EESC reached maximum concentrations around 1996 in mid-latitudes and around 2000 over polar regions. The 4 year difference is due to longer transport times to the polar stratosphere (see Figure 1-22 in Carpenter and Reimann [2014]).

Since EESC is expected to continue to decline, stratospheric ozone is expected to recover from the influence of ODSs. To provide policy-relevant statements on expected ozone recovery, the return of ozone concentrations to levels typical of 1980 has frequently been used as a benchmark [e.g., Bodeker and Waugh, 2007]. By comparing ozone concentrations projected by models with their simulated 1980 values, dates of return to 1980 levels can be identified. For Antarctica, October-mean total column ozone is projected to reach 1980 values between 2046 and 2057, about 5 years earlier than Cly will return to 1980

values [Eyring et al., 2010; Bekki and Bodeker, 2011].

There is evidence that the return of ozone to pre-1980 values is not equivalent to a full recovery of ozone from the effects of ODSs. Calculations of EESC (e.g., Montzka and Fraser [2003]; Clerbaux and Cunnold [2007]) show a clear upward trend in EESC before 1980, with relatively small increases in the 1960s followed by steeper increases in the 1970s. By 1980, polar stratospheric EESC had reached about 50% of its peak level around 2000, indicating that considerable Antarctic chemical

ozone loss should have occurred before 1980 [Carpenter and Reimann, 2014].

There are indications from both ground-based total column ozone measurements [Farman et al., 1985] and Antarctic ozonesonde stations [Solomon et al., 2005] of ozone depletion prior to 1980, suggesting an early effect of ODSs. Austin et al. [2010] and SPARC CCMVal [2010] found pre-1980 Antarctic ozone depletion in CCMVal-2 simulations. Some of the CCMs simulated an increasing Antarctic ozone mass deficit before 1980 and later return dates of Antarctic ozone to 1960 compared

to 1980. However, these studies did not provide any detail how much of the long-term ozone loss to 2000 had already occurred before 1980. In a more recent single-model CCM study using specified dynamics, Shepherd et al. [2014] found that 40 % of the long-term non-volcanic ozone loss occurred before 1980.

The purpose of this study is to quantify the extent to which Antarctic ozone was already affected by chemical ozone depletion in the period before 1980. The results provide a context for the utility of the return to 1980 levels as a benchmark of the degree

of recovery from the effects of ODSs. A multiple linear regression model is used to attribute simulated ozone changes in a set of 17 multi-decadal CCM simulations to changes in EESC. Importantly, non-linear dependence of ozone on EESC is included in the model so that a false positive does not arise as would be the case if pure linear dependence of ozone depletion on EESC had been assumed, i.e. a regression model with only linear dependence of ozone on EESC applied to an ozone signal that is constant from 1960 to 1980 and declines after 1980, would suggest EESC-induced ozone loss prior to 1980. The CCM

simulations were carried out in phase 2 of the SPARC (Stratosphere-troposphere Processes And their Role in Climate) Chemistry-Climate Model Validation (CCMVal-2) initiative [SPARC CCMVal, 2010]. These simulations are well suited for this study for two reasons: first, all models used emissions originating from the same prescribed scenario according to the CCMVal recommendations for REF-B1 simulations [Eyring et al., 2008]. This ensures that all models simulate the same surface mixing ratios of halocarbons [WMO, 2007] which then determine long-term stratospheric ozone depletion. Second,

the participating CCMs, due to their different horizontal and vertical resolutions, as well as varying degrees of implemented physical parameterizations, develop different dynamical variability and trends, in particular during winter and spring. As a result, the combination of identical prescribed ODS emissions and quite different dynamical variability simulated in the CCMs provides a rich set of simulations that can aid understanding of the observations. A detectable and consistent EESC-induced Antarctic ozone depletion prior to 1980 throughout the models would inform the use of 1980 as a benchmark for the definition of polar ozone recovery from ODSs.

The application of the least squares regression to separate the effects of halogens and temperature fluctuations on ozone changes is described together with the CCM output in Section 2. A comparison of the halogen-induced Antarctic ozone depletion in the models between 1960 and 2000 is compared with the derived ozone loss prior to 1980 in Section 3. A summary of the results, and conclusions, follow in Section 4.

## 2 Method, Models and Observations

To attribute CCM simulated Antarctic total column ozone changes to changes in stratospheric halogen loading and temperature variability, a multiple linear regression model is fitted to the total column ozone time series as:

$$O_3 = a\,ESC^2 + b\,ESC + c\,T' + d + \varepsilon \tag{1}$$

where a, b, c, and d are fit coefficients derived by least squares fitting of the equation to modelled or measured September to November total column ozone ($O_3$). Note that in contrast to the real atmosphere where EESC is derived from surface emissions of halocarbons, we use here equivalent stratospheric chlorine (ESC) at 50 hPa, i.e. the actual $Cl_y$ and $Br_y$ levels provided by the chemistry codes of the models, using a 60-fold efficacy for ozone destruction by bromine [Daniel et al., 1999]. $T'$ is the temperature anomaly at 100 hPa (with the climatological annual cycle subtracted). A constant offset is included, leading to the fit coefficient d, and $\varepsilon$ is the residual. All variables are averaged over Antarctica (60-90°S) and over Southern Hemisphere late winter/spring (September to November, SON). The regression model includes basis functions that account for the longer-term effects of halogens on ozone depletion ($ESC^2$ and ESC) as well as ozone variations due to year-to-year variations in mid-latitude planetary wave activity (represented by the corresponding Antarctic mean temperature anomalies T; e.g., Newman et al. [2004]; Huck et al. [2005]). The quadratic term ($ESC^2$) allows for non-linear dependence of ozone depletion on ESC related to the catalytic ClO destruction cycle [Jiang et al., 1996]. Moreover, it ensures that the regression model implicitly allows for constant ozone from 1960 to 1980. Tests with regression models including additional basis functions, such as the QBO, solar and volcanic effects, following the regression model used in SPARC-CCMVal [2010], resulted generally in larger unexplained residuals of the ozone time series than the simple model with a quadratic EESC fit used here (not shown). This indicates that temperature anomalies in the lower Antarctic stratosphere explain much if not all of the wintertime dynamical variability in total column ozone. Once the fit coefficients a, b, c, and d are determined for each of the CCMs, the degree of halogen-induced ozone depletion in any year, for each of the CCMs, can be derived using:

$$O_3 = a\,ESC^2 + b\,ESC \qquad\qquad\qquad\qquad\qquad\qquad\qquad\qquad\qquad\qquad (2)$$

Monthly mean total column ozone, temperature, $Cl_y$ and $Br_y$ concentrations from 17 CCMVal-2 simulations (see Table 1) were
analysed for the period 1960 to 2000 using the regression model. All simulations used observed transient forcings of ODSs,
greenhouse gas and sea surface temperatures/sea-ice concentrations as prescribed for the REF-B1 scenario by the CCMVal-2
initiative [Eyring et al., 2008]. Some models within the set included some sources of natural variability such as background
and volcanic aerosol, solar variability, the Quasi-Biennial-Oscillation (QBO), and ozone and aerosol precursors, while others
did not. The inclusion or exclusion of these factors was found to have no effect on the results of our study.

For comparison with observations, the same regression model was applied to a 1979 to 2000 database of total ozone column
measurements derived by combining measurements from multiple space-based instruments corrected for offsets and drifts
against the ground-based Dobson and Brewer spectrophotometer networks [Bodeker et al., 2005]. The data set combines total
column ozone measurements from Total Ozone Mapping Spectrometer (TOMS) instruments, the Global Ozone Monitoring
Experiment (GOME), Solar Backscatter Ultra-Violet (SBUV) instruments and the Ozone Monitoring Instrument (OMI). A
monthly mean, polar cap mean (60-90°S), total column ozone time series was calculated from this database. For the pre-
satellite era (before 1979), total column ozone measurements from Brewer and Dobson spectrophotometers at four Antarctic
stations (Faraday (previously Argentine Islands) at 65.3°S since 1957; Halley at 73.5°S since 1957; Syowa at 69°S since 1961;
and South Pole at 90°S since 1961) were used to estimate Antarctic mean (60°S-90°S) total column ozone. First the monthly
mean time series of total column ozone measurements at Argentine Islands/Faraday was combined with the time series of
measurements from Syowa to create a single time series representative of ozone changes on the periphery of the continent.
Systematic differences between Argentine Islands/Faraday and Syowa, arising primarily from their different locations, were
accounted for by averaging differences between temporally coincident monthly means (Argentine Islands/Faraday 12.58 DU
higher than Syowa on average). Whether Argentine Islands/Faraday is corrected against Syowa or vice versa is irrelevant as
the combined Argentine Islands/Faraday and Syowa time series is simply used as a predictor in a regression model and is
therefore insensitive to their absolute value. Monthly means were calculated from daily data where measurements made outside
of the circumpolar vortex (diagnosed from 550 K potential vorticity fields) were excluded from the calculation, and were
corrected in each year for temporal sampling bias. The three resultant location-specific monthly mean time series were then
used as basis functions in a regression model which was trained on available polar cap mean total column ozone obtained from
the observational database described above. Once trained, the regression model can be used to generate estimates of monthly
mean polar cap total column ozone from available monthly means at the ground-based measurement sites. Different forms of
the regression model were constructed depending on which location time series had missing data. This approach generates
robust estimates in the pre-satellite era in a way that introduces as little additional information as possible, errs on the side of
under-estimating the variability rather than over-estimating the variability, avoids spatial interpolation, and avoids the use of

ancillary data such as output from CCMs. Antarctic mean temperatures were derived from NCEP/NCAR reanalyses [Kalnay et al., 1996]. The EESC time series was taken from Newman et al. [2007], assuming a mean transport time of 5 years. Uncertainty values of the total column ozone and further derived quantities have been calculated by applying the standard formulae for error propagation to determine the uncertainties on the regression model fit coefficients.

## 3 Halogen-Induced Antarctic Ozone Loss

Figure 1 shows the time series of the SON average ESC at 50 hPa over Antarctica from 1960 to 2000, simulated by the CCMs used in this study. Also included are the EESC time series derived from measurements of halogen containing substances following the method of Newman et al. [2007] for mean transit times of 4 and 5 years. To facilitate comparison between different CCMs, the ESC values of the individual CCMs have been adjusted to a common baseline by subtracting their individual 1960 values and then adding the multi-model mean value for 1960. The same adjustment was applied to the EESC time series. All models show a slow ESC increase in the 1960s and 1970s, followed by a steeper increase until the mid to late 1990s. Except for one CCM, which simulates an ESC increase of 3.7 ppb between 1960 and 2000, the majority of the models simulate an ESC increase of about 2.8 ppb from 1960 to 2000. This ensemble shows good agreement with the EESC time series that is based on a 5-year transport time representative for polar latitudes [e.g., Newman et al., 2007]. A few models show a flattening of Antarctic ESC in the second half of the 1990s, and reach a smaller average ESC increase of about 2.5 ppb. The evolution of polar ESC in these models is more similar to the EESC time series with a 4-year transport time to middle and higher latitudes. This indicates that these models have a faster transport of constituents towards polar latitudes than observed. It is evident that elevated ESC abundances appeared in the Antarctic lower stratosphere before 1980. The simulated increase of about 0.9 ppb from 1960 to 1980 corresponds to one third of the increase between 1960 and 2000.

## 3.1 Observed Ozone Loss

Figure 2 shows the observed Antarctic total column ozone time series between 1960 and 2000 (black line, left panel). The observations include estimates of polar cap total column ozone, based on the four sites listed earlier, to 1978, and then satellite-based measurements thereafter. Year-to-year variations in SON mean Antarctic stratospheric ozone are apparent which fluctuate around a longer-term downward trend. The regression model fit to the observations (Equation 1, blue line in Figure 2, left panel) reproduces this behaviour well, explaining ~91 % of the variance in the observations. The residuals are normally distributed (Figure 2, right panel), adding confidence to the robustness of the regression model fit.

The Antarctic SON mean total column ozone evolution attributable to changes in EESC, as derived using Equation 2, is shown by the red line in Figure 2. The regression analysis quantifies an EESC-induced Antarctic total column ozone loss from 1960 to 1980 of $76.7 \pm 3.1$ DU, and from 1960 to 2000 of $136.0 \pm 10.9$ DU. Slightly more than half ($56.4 \pm 6.8$ %) of the Antarctic ozone loss caused by EESC in the period from 1960 to 2000 occurred already before 1980. While the EESC induced ozone change leads to a continuous downward trend of Antarctic column ozone through the 1960s and 1970s, the ozone time series itself shows a flatter evolution over the two decades. This can be explained by a warming of the southern polar lower

stratosphere in that period, associated with noticeable dynamically induced year-to-year temperature variations (Labitzke and Kunze [2005]; Newman and Rex [2007]). Antarctic winters with strong and cold polar vortices were alternating with others that developed dynamically disturbed polar vortices. The high total column ozone in Antarctic spring 1968 (Fig. 2, left), for example, was associated with a weak and warm polar vortex that broke down in early spring (see Fig. 6 in Langematz and

Kunze [2006]), while the low ozone column amounts in the springs 1966 and 1969 were connected to strong and cold vortices with late breakdown dates.

## 3.2 Ozone Loss in CCMs

As for the observations, simulated halogen-induced ozone losses in the CCMs were derived using the statistical model of Section 2. The blue line in Figure 3 (left panel) shows for example the results of fitting the full regression model (Equation 1)

to EMAC-FUB total column ozone (black line). The regression model, which explains 96 % of the variance in the CCM simulated ozone signal, reproduces well the slow downward trend and the year-to-year variability in the simulated Antarctic ozone. For all other CCMs (except for one outlier) similar regression results were obtained with the regression model explaining between 93 % and 99 % of the variance in the CCM signals.

By applying Equation 2, the ESC-induced Antarctic ozone loss in the CCMs was calculated. In EMAC-FUB, the regression

analysis suggests an ESC-induced Antarctic ozone loss from 1960 to 1980 of 39.8 ± 1.0 DU, and from 1960 to 2000 of 89.4 ± 4.2 DU (red line in Figure 3). 44.5 ± 3.2 % of the ozone loss caused by ESC in the period from 1960 to 2000 occurred before 1980.

Table 2 lists the ESC-induced SON Antarctic ozone losses for all CCMs used in this study, together with the observational estimate. The ESC-only terms in the regression model have been used to separately diagnose ozone losses for the periods

1960-1980 and 1960-2000; the percentage contribution of the 1960-1980 ozone loss to the ozone loss over the whole period 1960-2000 is given in the rightmost column. As expected, all CCMs simulate ESC-induced Antarctic ozone depletion in late winter and spring between 1960 and 2000, ranging from 53.9 ± 4.1 DU to 182.0 ± 15.5 DU. The observed EESC-induced ozone decline over that period amounts to 136.0 ± 10.9 DU.

The temporal evolution of SON Antarctic mean total column ozone from 1960 to 2000 for the individual CCMs and

observations is illustrated in Figure 4. Absolute values have been adjusted to a common baseline, i.e. the mean total column ozone in 1960 of all CCMs. It is evident that ozone depletion by halogens started prior to 1980. All CCMs consistently simulate an ESC-induced decrease in SON mean Antarctic total column ozone of between 19.9 ± 1.0 DU and 90.7 ± 3.5 DU from 1960 to 1980. The ESC-induced ozone loss is, however, masked in those CCMs that reproduce the observed polar stratospheric warming between 1960 and 1980, and is enhanced in those CCMs that simulate a cooling in that period. Figure 5 shows the

evolution of the polar cap mean SON mean temperature at 100 hPa between 1960 and 2000, fitted with piecewise linear trends from 1960 to 1980 and from 1980 to 2000. While the Antarctic lower stratosphere temperature observations showed warming in SON from 1960 to 1980, the CCMs span a wide range of trends, indicating different temperature trend regimes resulting from model dynamical variability differing from what happened in reality. In addition, the absolute temperature values differ

between the CCMs by more than 10 K, which directly affects the potential for chemical ozone loss in the models. Figure 6 compares linear trends in Antarctic total column ozone against polar 100 hPa temperatures in the CCMs and observations between 1960 and 1980. CCMs simulating a stronger cooling of the Antarctic lower stratosphere, and therefore a more stable winter-time polar vortex, show a stronger total column ozone decline than those models that produce no cooling or even a
warming.

According to the CCM simulations, ESC caused Antarctic ozone depletion prior to 1980 of $26.4 \pm 3.4$ to $49.8 \pm 6.2$ % of the total depletion between 1960 and 2000. This estimate is marginally lower than the observed value of $56.4 \pm 6.8$ % with only a few CCMs replicating the observed pre-1980 Antarctic ozone depletion within the uncertainties. Most of the ESC-induced Antarctic ozone depletion between 1960 and 1980 took place in the second decade of this period. For the observations, the
regression yields an ozone depletion of $-41.9 \pm 2.7$ DU between 1970 and 1980; $54.6 \pm 1.3$ % of the ozone loss between 1960 and 1980 took place in the second decade of this period. With the exception of 3 CCMs, the models generally show a stronger ESC-induced ozone loss between 1970 and 1980 of the order of 60 to 75 % after 1970, in agreement with elevated ESC in the CCMs after 1970 (Figure 1).

## 4 Discussion and Conclusion

Output from CCMVal-2 REF-B1 CCM simulations forced by a realistic transient scenario of ODSs for the period 1960 to 2000 was used to investigate anthropogenic ozone depletion over Antarctica prior to 1980. A regression model was fitted to Antarctic SON vortex average total ozone columns taking effects of ESC and temperature variations into account. The regression results with regression coefficients varying from $R^2=0.89$ to $R^2=0.99$ showed that Antarctic ozone levels are dominated by halogen chemistry and dynamical effects. By evaluating only the ESC terms in the regression model, we were
able to derive ESC-induced ozone depletion for the periods 1960-1980 and 1960-2000.

The observed decrease in total column ozone between 1960 and 2000 was reproduced - within its uncertainty range - by 7 models (CMAM, LMDZrepro, UMSLIMCAT, UMUKCA-METO, UMUKCA-UCAM, WACCM and ULAQ). Two of these CCMs (CMAM, WACCM) obtained the highest ranking in an evaluation of their photochemistry and transport characteristics performed within the SPARC CCMVal activity [SPARC CCMVal, 2010] and discussed in Chapter 2 of the 2010 WMO ozone
assessment [WMO, 2011], providing confidence in the robustness of their results. 4 CCMs (AMTRAC3, CNRM-ACM, GEOSCCM, MRI) simulated a stronger ozone decline, and 6 CCMs (CAM3.5, CCSRNIES, EMAC, EMAC-FUB, NIWA-SOCOL, SOCOL) underestimated the observed ozone decline. This divergent model behaviour may be due to the representation of polar ozone chemistry in the models, their dynamical and transport characteristics, or to a combination of both. Based on the detailed evaluation performed as part of the SPARC CCMVal activity [SPARC CCMVal, 2010], we found
in our study that the CCMs that represent the observations well, generally (with one exception) show a good potential for chlorine activation and (all) a good representation of chemical ozone depletion in Antarctic spring. CCMs with a stronger ozone loss than observed (cf. Table 2) partly tend to a slight overestimation of chemical ozone depletion (AMTRAC3, GEOSCCM). For some CCMs with weaker ozone decline between 1960 and 2000 a consistent underestimation of chemical

ozone depletion was found (CCSRNIES, EMAC, CAM3.5). Thus, the deviations of some CCMs from the observed ozone decline can partly be explained by deficiencies in their polar ozone chemistry. However, in addition, models that underestimate the observed ozone decline were found to suffer from either a too fast transport of air into the Antarctic polar vortex (SOCOL, NIWA-SOCOL) or a too weak insolation of the polar vortex from mid-latitudes in the lower stratosphere (CAM3.5,

CCSRNIES, EMAC, SOCOL, NIWA-SOCOL). Both effects lead to lower ESC concentrations by the end of the 20th century in these models (cf. Fig. 1), and as a result an underestimation of the observed polar ozone decline due to ESC.

Consistent negative Antarctic ozone changes were diagnosed in the CCMs prior to 1980 as a result of chemical depletion by ESC. This pre-1980 halogen-induced Antarctic ozone depletion amounts to values between $26.4 \pm 3.4$ and $49.8 \pm 6.2$ % of the simulated ozone depletion between 1960 and 2000. Hence the CCM simulations are consistent with the observational estimate

of a significant EESC induced ozone decline in 1960-1980, albeit nearly all CCMs underestimate the observed decline of 56.4 $\pm$ 6.8 %, derived from the NIWA combined total column ozone data base. However, note that the two CCMs, ranked highest in the SPARC CCMVal evaluation of their photochemistry and transport characteristics, CMAM and WACCM, [SPARCCCMVal, 2010] nearly agree with the observed decline between 1960 and 1980 within its uncertainty range.

Apart from the chemical and dynamical performance of the models, their underestimation of the ozone loss might be related

to a sampling bias of the CCMs that include high latitudes with little ozone depletion in their polar cap mean, while the observations are made in sunlight. However, this bias exists in September only. Moreover, the region of the polar cap that is in perpetual darkness is very small (as a fraction of the whole area poleward of 60°S) and shrinks to zero by the end of September. So, this effect should be of the order of a few percent. Another potential reason for the underestimation of the Antarctic ozone decline before 1980 in most CCMs is the effect on chemical ozone depletion by short-lived bromine

compounds of natural biogenic origin, so-called very short lived substances (VSLS). The effects of VSLS which contribute 20-30 % to the present-day stratospheric bromine content [WMO, 2011] on Antarctic stratospheric ozone were not included in the CCMVal-2 simulations. Braesicke et al. [2013] and Sinnhuber and Meul [2014] showed that taking brominated VSLS in their CCMs into account leads to a significant reduction of Antarctic polar ozone. In a transient REF-B1 simulation using the same FUB-EMAC CCM as included in this study but with prescribed VSLS sources, Sinnhuber and Meul [2014] found a

reduction of October mean ozone in the lower Antarctic stratosphere of more than 20 %. However, although constant VSLS emissions were prescribed over the whole simulation period of 1960-2005, the impact of VSLS was stronger in the most recent period after 1980 with enhanced chlorine due to combined bromine-chlorine catalytic ozone loss cycles. Hence, including the VSLS effect leads to an enhancement of the 1960-2000 Antarctic ozone depletion, but reduces the relative change in 1960-1980 compared to the full period. Further insight is expected from the analysis of the new CCMI- simulations that will include

the effects of VSLS.

Our results show that CCM modelled declines in Antarctic polar cap average total column ozone from 1960 to 1980 are not intrinsically in disagreement with observations which show little change in polar cap average ozone over this period. The apparent discrepancy results from the particular instance of reality in which Antarctic stratospheric temperatures increased over the period 1960 to 1980 that significantly offset the EESC-induced depletion of ozone. In the CCM simulations in which

stratospheric warming occurs from 1960 to 1980, similar to observations, no statistically significant changes in ozone prior to 1980 are observed. These results reiterate that while the return of Antarctic ozone to 1980 levels remains a valid milestone on the path to recovery, attaining this milestone cannot be indicative of the full recovery of Antarctic ozone from the effects of ODSs since appreciable ODS-induced ozone depletion occurred prior to 1980.

## 5 Data Availability

The CCM data used in this study are available from the CCMVal-II database at the British Atmospheric Data Centre (British Atmospheric Data Centre, 2009) (http://browse.ceda.ac.uk/browse/badc/ccmval/data/CCMVal-2/Reference_Runs/REF-B1/). The NIWA combined total column ozone database can be obtained from http://www.bodekerscientific.com/data/the-bdbp. NCEP/NCAR reanalyses are available from http://rda.ucar.edu/datasets/ds090.0/#description.

*Acknowledgements*. We acknowledge the modelling groups for making their simulations available for this analysis, the Chemistry-Climate Model Validation (CCMVal) Activity of WCRP's (World Climate Research Programme) SPARC (Stratosphere-troposphere Processes And their Role in Climate) project for organizing and coordinating the model data analysis activity, and the British Atmospheric Data Centre (BADC) for collecting and archiving the CCMVal model output. We thank Ted Shepherd for his thoughts on the methodology underlying this paper. Thank you to Christian Blume for input to Figure 1. UL was supported by the International Bureau of the Federal Ministry of Education and Research.

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

| Model | Group |
|-------|-------|
| AMTRAC3 | GFDL, USA |
| CAM3.5 | NCAR, USA |
| CCSRNIES | NIES, Japan |
| CMAM | Univ. of Toronto and York Univ., Canada |
| CNRM-ACM | Météo-France, France |
| EMAC | MPI Chemistry Mainz, Germany |
| EMAC-FUB | Freie Universität Berlin, Germany |
| GEOSCCM | NASA/GSFC, USA |
| LMDZrepro | IPSL, France |
| MRI | Meteorological Research Institute, Japan |
| NiwaSOCOL | NIWA, New Zealand |
| SOCOL | PMOD/WRC and ETHZ, Switzerland |
| ULAQ | University of L'Aquila, Italy |
| UMSLIMCAT | University of Leeds, UK |
| UMUKCA-METO | MetOffice, UK |
| UMUKCA-UCAM | University of Cambridge, UK |
| WACCM | NCAR, USA |

**Table 1: List of participating CCMs.**

| Model | 1960 to 1980 | 1960 to 2000 | 1960 to 1980 |
|-------|--------------|--------------|--------------|
|  | (DU) | (DU) | (% of 1960 to 2000) |
| AMTRAC3 | $-68.4 \pm 1.6$ | $-165.3 \pm 6.7$ | $41.4 \pm 2.6$ |
| CAM3.5 | $-35.6 \pm 2.0$ | $-84.5 \pm 7.9$ | $42.1 \pm 6.2$ |
| CCSRNIES | $-37.6 \pm 1.9$ | $-91.8 \pm 13.1$ | $41.0 \pm 7.9$ |
| CMAM | $-63.5 \pm 1.2$ | $-140.4 \pm 4.7$ | $45.2 \pm 2.4$ |
| CNRM-ACM | $-90.7 \pm 3.5$ | $-182.0 \pm 15.5$ | $49.8 \pm 6.2$ |
| EMAC | $-19.9 \pm 1.0$ | $-53.9 \pm 4.1$ | $36.9 \pm 4.8$ |
| EMAC-FUB | $-39.8 \pm 1.0$ | $-89.4 \pm 4.2$ | $44.5 \pm 3.2$ |
| GEOSCCM | $-57.0 \pm 1.4$ | $-170.8 \pm 6.1$ | $33.4 \pm 2.0$ |
| LMDZrepro | $-66.8 \pm 1.3$ | $-142.4 \pm 4.4$ | $46.9 \pm 2.4$ |
| MRI | $-75.2 \pm 1.0$ | $-166.2 \pm 3.0$ | $45.2 \pm 1.4$ |
| NiwaSOCOL | $-41.2 \pm 0.0$ | $-93.7 \pm 6.6$ | $44.0 \pm 3.1$ |
| SOCOL | $-28.6 \pm 0.2$ | $-80.7 \pm 4.7$ | $35.4 \pm 2.4$ |
| ULAQ | $-43.9 \pm 6.1$ | $-149.3 \pm 34.3$ | $29.4 \pm 10.8$ |
| UMSLIMCAT | $-59.4 \pm 4.3$ | $-135.5 \pm 15.1$ | $43.8 \pm 8.1$ |
| UMUKCA-METO | $-32.8 \pm 1.6$ | $-124.0 \pm 9.6$ | $26.4 \pm 3.4$ |
| UMUKCA-UCAM | $-39.6 \pm 1.3$ | $-130.8 \pm 6.5$ | $30.3 \pm 2.5$ |
| WACCM | $-58.9 \pm 2.3$ | $-130.8 \pm 10.2$ | $45.0 \pm 5.3$ |
| **OBS** | **$-76.7 \pm 3.1$** | **$-136.0 \pm 10.9$** | **$56.4 \pm 6.8$** |

**Table 2: Halogen-induced Antarctic ozone depletion in DU in CCM simulations and observations for the periods: 1960 to 1980 and 1960 to 2000. 1σ-uncertainties were derived by applying error propagation rules. The rightmost column indicates changes from 1960 to 1980 as percentage of the changes from 1960 to 2000.**

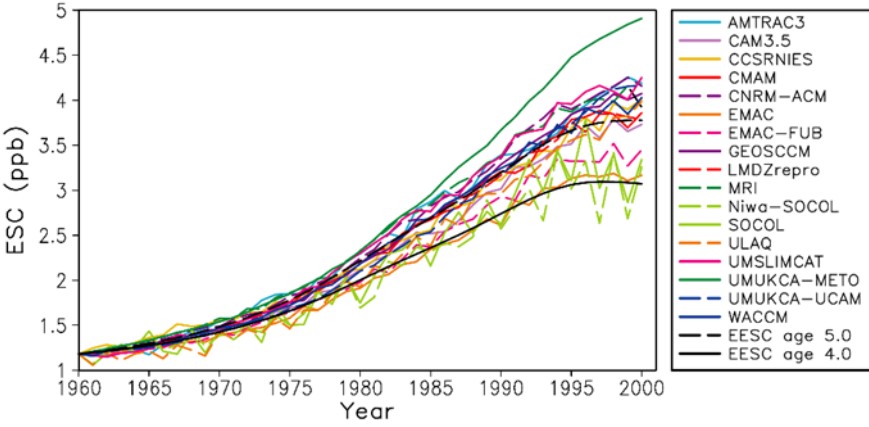

**Figure 1: Evolution of Antarctic SON average ESC (in ppb) in the CCMs between 1960 and 2000, adjusted to a common baseline of 1960. Black lines show EESC (in ppb), provided by Newman et al. [2007] for mean transit times of 4 (solid) or 5 (dashed) years.**

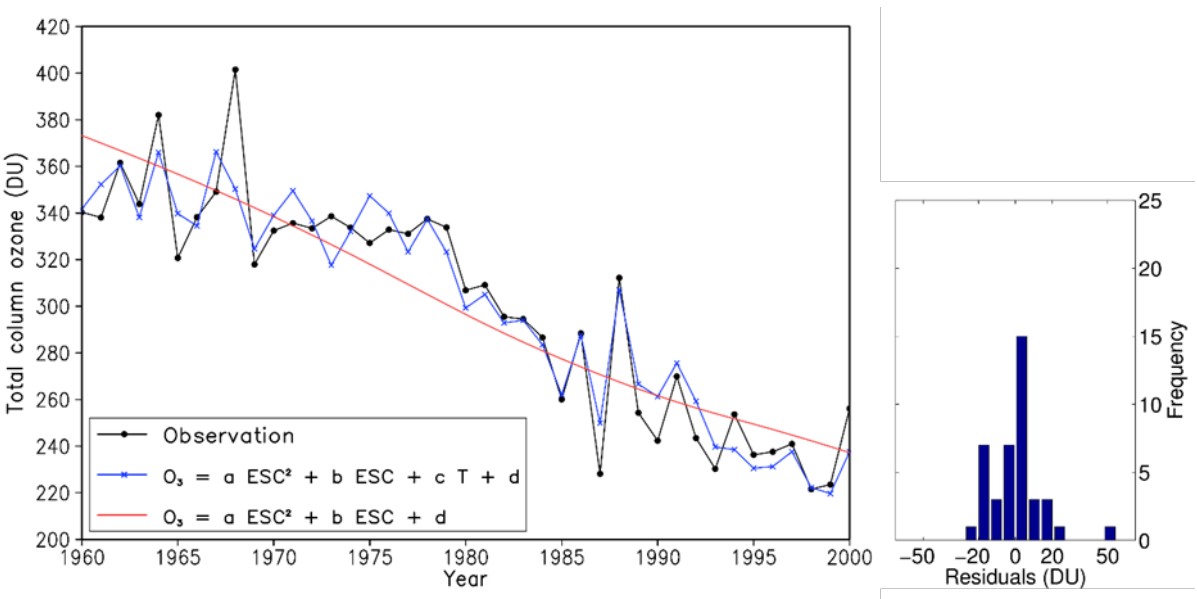

**Figure 2: Left panel: Antarctic mean, SON mean total column ozone from observations (black line), from the full regression model (blue line), and from the regression model including the EESC term only (red line). Right panel: Histogram of the residuals.**

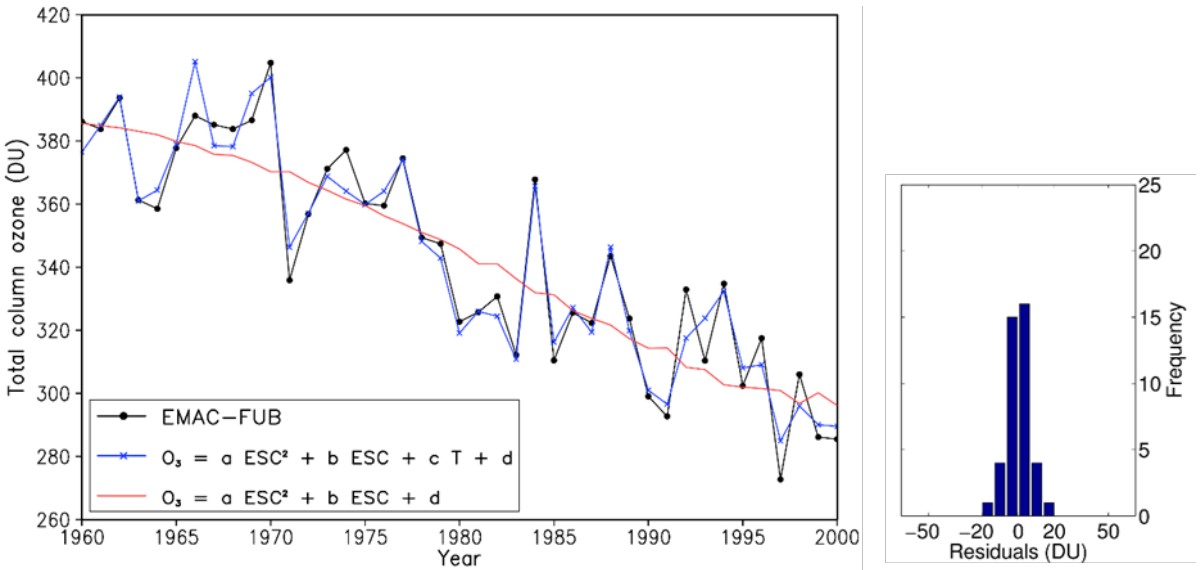

**Figure 3: Left panel: Total column ozone from the EMAC-FUB CCM (black line), from the full regression model (blue line), and from the regression model including the ESC term only (red line). Right panel: Histogram of the residuals.**

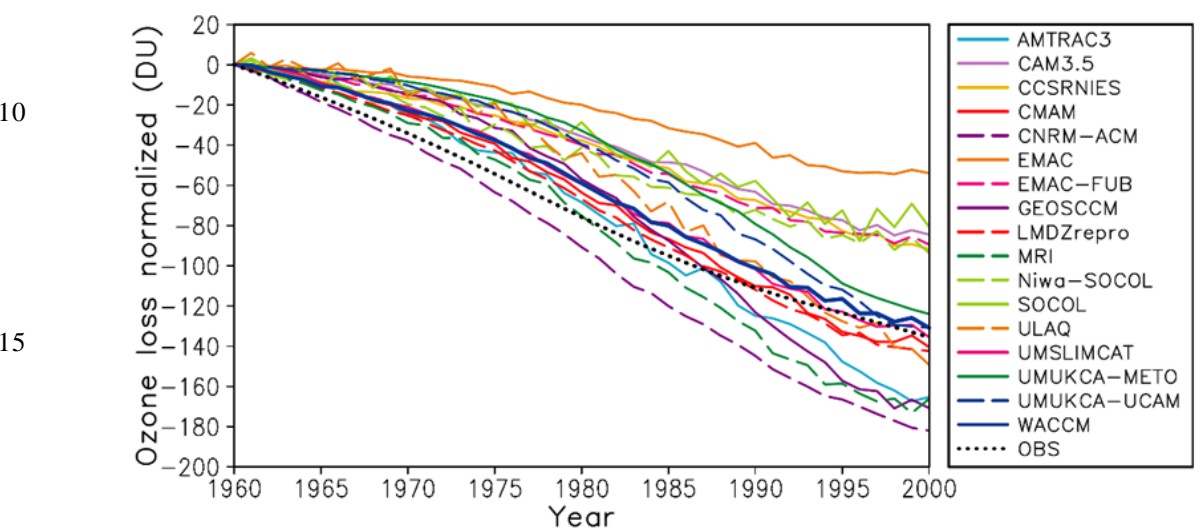

20 **Figure 4: Antarctic total ozone column depletion (in DU, September/October/November average) between 1960 and 2000 due to ESC in REF-B1 CCM simulations and due to EESC in observations, adjusted to a common baseline (1960 mean of CCMs).**

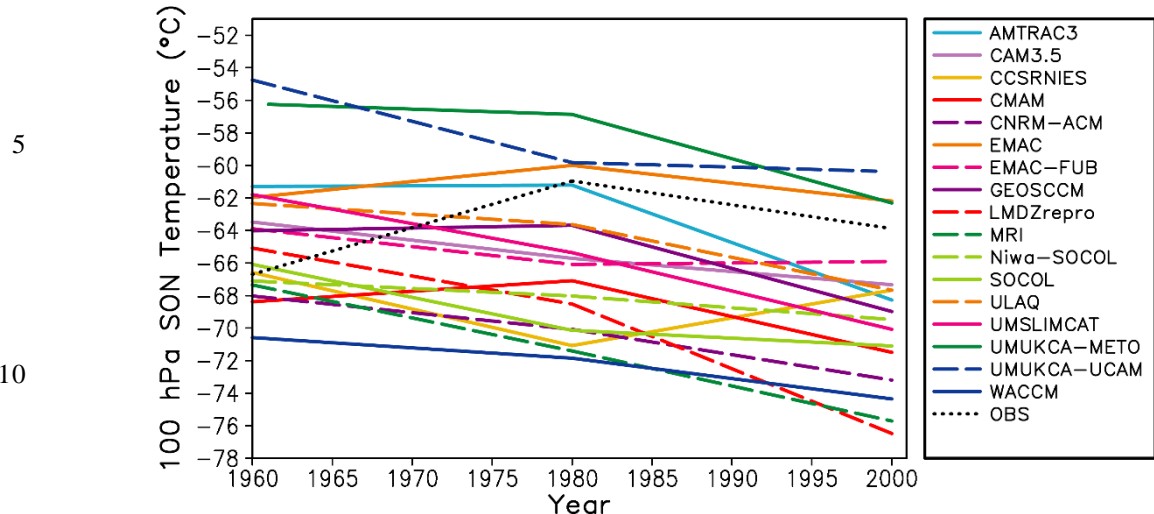

Figure 5: Antarctic temperature in 100 hPa (in °C, September/October/November average) between 1960 and 2000 in REF-B1 CCM simulations (coloured lines) and observations (black). Temperatures have been constructed by combining the offset and linear trend coefficients of a regression model similar to equation 1 applied piecewise to two periods from 1960 to 1980 and 1980 to 2000

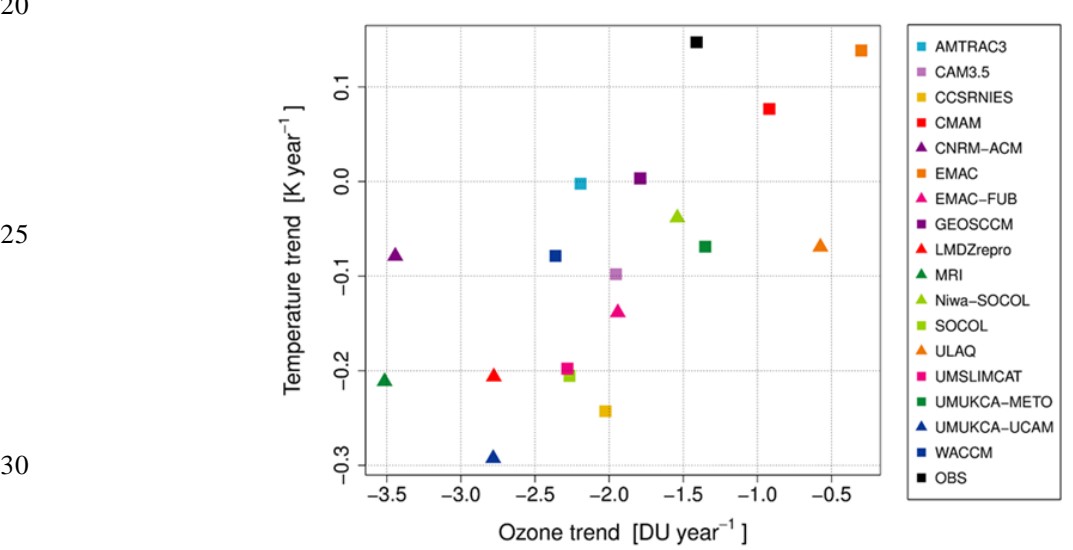

Figure 6: Scatter diagram of the linear trends in the Antarctic total column ozone (in DU/year) and the Antarctic temperature at 100 hPa (in K/year) in September/October/November between 1960 and 1980 for the 17 CCMs of this study and the observations.