# Peer review of "Antarctic Ozone Depletion between 1960 and 1980 in Observations and Chemistry-Climate Model Simulations"

_Atmospheric Chemistry and Physics, 2016_

## Referee Comment (RC1) · Anonymous Referee #2 · 30 Sep 2016

This paper quantifies the amount of ozone loss that happened prior to 1980 using several chemistry-climate models. The 1980 return level is a widely used and policy-relevant metric, but these results show comprehensively that while this metric is useful it does not give a good indication of complete stratospheric ozone recovery. Overall the paper is well written and structured. Below are some comments that could be addressed to further improve the paper.

General comments:

1. It would be very interesting to run a similar analysis on the newer CCMI (Chemistry-Climate Modelling Initiative) simulations. This probably wouldn't change the main conclusions of the paper, but it might be good to use some newer simulations. One could

even compare the results of the CCMVal simulations with the CCMI models to investigate whether the differences between the two ensembles are smaller/larger.

2. P3L2: Why specify 'stratospheric winter' is it different from tropospheric winter? Is it specifically the Southern Hemisphere winter?

3. P4L19: How were the systematic differences between Syowa and Faraday corrected? Perhaps a sentence or two about this might be useful.

4. P5L14-16: How do the ground-based measurements compare to the satellite observations post 1978? Figure 2 shows just one line for both – how were they linked to form one time series? Were the satellite data averaged over the entire 60-90°S region?

5. P6L15-17: Why is there such a large range in the model simulated ESC-induced ozone loss (min 54DU and max 182DU from 1960-2000 (even more extreme differences between models pre-1980))? Is this a result of the different dynamics between models? Or are there differences in the chemistry?

6. P6L28-30: Why do so few models show an ozone loss within the observational uncertainties? Is there a reason that so many of them underestimate the loss? (i.e. is there a particular bias that needs to be addressed?)

Minor technical issues:

P1L29: Wuebbels -> Wuebbles

P4L6: Please spell the acronym 'GHG' out.

P4L23: and were and corrected -> and were corrected.

P4L16: in 65.3°S -> at 65.3°S (as well as the other latitude specifications in this line and the next).

P6L1: 3.1 -> 3.2

P6L3: 'shows as an example the results of' -> 'shows an example of the results of'

---

## Referee Comment (RC2) · Anonymous Referee #1 · 3 Oct 2016

General Comment: This paper examines the Antarctic ozone depletion between 1960 and 1980 in both observations and 17 Chemistry Climate Models (CCMs) based on the REF-B1 scenario from CCMVal2. These models derive an anthropogenic depletion from 1960-1989 between 26.4% and 49.8% of the total period (1960-2000). Observations over the same period suggest a higher depletion of 56%. The paper is clearly written, concise, and adds to the scientific understanding of what the return date choice for "full recovery" implies. I recommend this paper be published after minor revisions (see below).

Specific comments.

Page 3 lines 3-4 All the models certainly do show a 1960-1980 depletion (26.4-49.8%),

with approximately six model showing values less than ∼ 35%. For these 17 models there was large effort to understand how well these models represented transport, dynamics, and chemistry (i.e., the SPARC Report on the Evaluation of Chemistry-Climate Models, 2010). That is, the range in models is not just due to different dynamical variability. It would be interesting to highlight the models that did better in these process oriented diagnostics in Table 2. This was the approach used in Chapter 2 of the 2010 WMO assessment.

You suggest the the temperature trend in the 1960-1980 period was different in observations relative to most models (Figure 5). Are there any other issues with the models that could explain the lower depletion in this period? E.g., the CCMs used in this study also did not include additional very-short lived bromine (VSL) species. This addition 5-7 pptv of inorganic bromine should contribute to the underestimate the total loss in the 1960-1980 period. It will be interesting if you (not for this paper) redo this analysis for the CCMI models that include this additional VSL bromine source.

Page 3, equation 1. The authors did a very nice job of explaining the approach of determining the degree of halogen-induced ozone for the 1960-1980 period. Question: the temperature anomaly at 100hPa is used in the regression fit to address dynamical variability. What equation (1) does not address is the sensitivity of the ozone chemistry in the model to absolute biases in temperature. It therefore would be very informative to show the lower polar stratosphere absolution temperature evolution similar to Figure 4. E.g., if two models both show a similar representation of ESC (i.e., consistent transport/dynamics) and a similar absolute temperature trend (new figure), and would happen to have a different temperature trend vs ozone trend sensitivity (Figure 5) – I believe this would highlight issues in the chemistry representation between the two models. Generally, it would be nice to comment on how this technique could be used to evaluate model components.

Page 4. The discussion of how the observations are combined are in reasonably detailed. Based on this discussion and use of equation 1 (and 2) this work suggests a

decline of 56.4 +-6.8%. Maybe I missed it, but how did you come up with an uncertainty value for this decline (i.e., +-6.8%)? Also, since the results are for SON mean total ozone polar cap average (e.g., Figure 2) – are you masking the model results for periods that are in the dark with little ozone depletion (e.g., high latitudes in September)? That is, are you treating the model and observations derivation of the 1960-1980 decline in a consistent manner?

Figure 2 caption is missing the prime symbol in "c T".
* * *

---

## Author Comment (AC1) · 7 Dec 2016

We would like to thank the reviewer for the time and the useful comments that helped to clarify some important aspects of the model results.

This paper quantifies the amount of ozone loss that happened prior to 1980 using several chemistry-climate models. The 1980 return level is a widely used and policy-relevant metric, but these results show comprehensively that while this metric is useful it does not give a good indication of complete stratospheric ozone recovery. Overall the paper is well written and structured. Below are some comments that could be addressed to further improve the paper.

General comments:

1. It would be very interesting to run a similar analysis on the newer CCMI (Chemistry-Climate Modelling Initiative) simulations. This probably wouldn't change the main conclusions of the paper, but it might be good to use some newer simulations. One could even compare the results of the CCMVal simulations with the CCMI models to investigate whether the differences between the two ensembles are smaller/larger.

We chose to use simulations from the CCMVal-2 database because it is a comprehensive and well-documented data set. For the discussion of the differences between the individual models, which has been added in the revised manuscript, we relied heavily on the SPARC-CCMVal report which provided valuable information on the chemical, dynamical and transport properties of the models used in our study. A corresponding analysis of the CCMI simulations has been announced as a CCMI project and will be performed once sufficient data will be available. However, because such an evaluation of the CCMI simulations is not yet available, we made a conscious decision not to use CCMI simulations in this study.

2. P3L2: Why specify 'stratospheric winter' is it different from tropospheric winter? Is it specifically the Southern Hemisphere winter?

'stratospheric' has been removed

3. P4L19: How were the systematic differences between Syowa and Faraday corrected? Perhaps a sentence or two about this might be useful.

On months where both Syowa and Faraday had valid monthly means, their differences were calculated. The average of those differences was 12.58 DU (Faraday higher). The following sentences have been added to the paper on P4, L20:

"*First the monthly mean time series of total column ozone measurements at Argentine Islands/Faraday was combined with the time series of measurements from Syowa to create a single time series representative of ozone changes on the periphery of the continent. Systematic differences between Argentine Islands/Faraday and Syowa, arising primarily from their different locations, were accounted for by averaging differences between temporally coincident monthly means (Argentine Islands/Faraday 12.58 DU higher than Syowa on average). Whether Argentine Islands/Faraday is corrected against Syowa or vice versa is irrelevant as the combined Argentine Islands/Faraday and Syowa time series is simply used as a predictor in a regression model and is therefore insensitive to their absolute value.*"

4. P5L14-16: How do the ground-based measurements compare to the satellite observations post 1978? Figure 2 shows just one line for both – how were they linked to form one time series? Were the satellite data averaged over the entire 60-90◦S region?

We have not shown comparisons of the ground-based measurements to the satellite observations post 1978 because it is not relevant to the paper. The ground-based measurements are never used, in isolation, in this analysis. The time series at the three locations:
1) Argentine Islands + Syowa
2) Halley
3) South Pole
are used, collectively, as predictors for monthly mean polar cap mean total column ozone. Because the ground-based observations are used only as basis functions in a regression model that relates those three time series to polar cap means, they could each be systematically different from the satellite data by 500 DU and it would make no difference at all to the pre-1979 polar cap mean time series created in this analyses. Therefore, we felt it unnecessary to explore systematic biases between the ground- and satellite-based measurements in this paper.

The pre-satellite era time series was simply spliced onto the front of the satellite-era time series. Because of the way in which the pre-satellite era time series was constructed, there is no systematic bias between the two time series.

This level of detail describing the construction of the pre-1979 time series was felt unnecessary and so has not been included in the paper.

Yes, the satellite data were averaged over the entire 60-90°S region. A sentence to this effect has been added to the paper (P4, L13):

"*The data set combines total column ozone measurements from Total Ozone Mapping Spectrometer (TOMS) instruments, the Global Ozone Monitoring Experiment (GOME), Solar Backscatter Ultra-Violet (SBUV) instruments and the Ozone Monitoring Instrument (OMI). A monthly mean, polar cap mean (60-90°S),....*"

5. P6L15-17: Why is there such a large range in the model simulated ESC-induced ozone loss (min 54DU and max 182DU from 1960-2000 (even more extreme differences between models pre-1980))? Is this a result of the different dynamics between models? Or are there differences in the chemistry?

The spread between the CCMs may partly be explained by differences in the chemical ozone depletion in the models (SPARC CCMVal, 2010). The large range is, however, particularly due to CCMs that considerably underestimate the observed ozone decline. These CCMs were shown to have a too fast transport of air into the polar vortex or too weak transport barriers between mid-latitudes and the polar vortex (SPARC CCMVal, 2010), both leading to lower ESC values by 2000 and a weaker ozone depletion than observed. We have now added the following paragraph to the 'Discussion and Summary' section with more detailed explanations of the results:

"*The observed decrease in total column ozone between 1960 and 2000 was reproduced - within its uncertainty range - by 7 models (CMAM, LMDZrepro, UMSLIMCAT, UMUKCA-METO, UMUKCA-UCAM, WACCM and ULAQ). Two of these CCMs (CMAM, WACCM) obtained the highest ranking in an evaluation of their photochemistry and transport characteristics performed within the SPARC CCMVal activity [SPARC CCMVal, 2010] and discussed in Chapter 2 of the 2010 WMO ozone assessment [WMO, 2011], providing confidence in the robustness of their results. 4 CCMs (AMTRAC3, CNRM-ACM, GEOSCCM, MRI) simulated a stronger ozone decline, and 6 CCMs (CAM3.5, CCSRNIES, EMAC, EMAC-FUB, NIWA-SOCOL, SOCOL) underestimated the observed ozone decline. This divergent model*

*behaviour may be due to the representation of polar ozone chemistry in the models, their dynamical and transport characteristics, or to a combination of both. Based on the detailed evaluation performed as part of the SPARC CCMVal activity [SPARC CCMVal, 2010], we found in our study that the CCMs that represent the observations well, generally (with one exception) show a good potential for chlorine activation and (all) a good representation of chemical ozone depletion in Antarctic spring. CCMs with a stronger ozone loss than observed (cf. Table 2) partly tend to a slight overestimation of chemical ozone depletion (AMTRAC3, GEOSCCM). For some CCMs with weaker ozone decline between 1960 and 2000 a consistent underestimation of chemical ozone depletion was found (CCSRNIES, EMAC, CAM3.5). Thus, the deviations of some CCMs from the observed ozone decline can partly be explained by deficiencies in their polar ozone chemistry. However, in addition, models that underestimate the observed ozone decline were found to suffer from either a too fast transport of air into the Antarctic polar vortex (SOCOL, NIWA-SOCOL) or a too weak insolation of the polar vortex from mid-latitudes in the lower stratosphere (CAM3.5, CCSRNIES, EMAC, SOCOL, NIWA-SOCOL). Both effects lead to lower ESC concentrations by the end of the 20th century in these models (cf. Fig. 1), and as a result an underestimation of the observed polar ozone decline due to ESC.*

*Consistent negative Antarctic ozone changes were diagnosed in the CCMs prior to 1980 as a result of chemical depletion by ESC. This pre-1980 halogen-induced Antarctic ozone depletion amounts to values between 26.4 ± 3.4 and 49.8 ± 6.2 % of the simulated ozone depletion between 1960 and 2000. Hence the CCM simulations are consistent with the observational estimate of a significant EESC induced ozone decline in 1960-1980, albeit nearly all CCMs underestimate the observed decline of 56.4 ± 6.8 %, derived from the NIWA combined total column ozone data base. However, note that the two CCMs, ranked highest in the SPARC CCMVal evaluation of their photochemistry and transport characteristics, CMAM and WACCM, [SPARCCCMVal, 2010] nearly agree with the observed decline between 1960 and 1980 within its uncertainty range."*

6. P6L28-30: Why do so few models show an ozone loss within the observational uncertainties? Is there a reason that so many of them underestimate the loss? (i.e. is there a particular bias that needs to be addressed?)

The 6 CCMs that underestimate the observed ozone loss between 1960 and 2000 mainly suffer from deficits in the dynamics and transport of air (see reply to 5.). This has now been elaborated on in more detail in the new paragraph in the 'Discussion and Summary' section.

Minor technical issues:

P1L29: Wuebbels -> Wuebbles

done

P4L6: Please spell the acronym 'GHG' out.

done

P4L23: and were and corrected -> and were corrected.

done

P4L16: in 65.3◦S -> at 65.3◦S (as well as the other latitude specifications in this line and the next).

done

P6L1: 3.1 -> 3.2

done

P6L3: 'shows as an example the results of' -> 'shows an example of the results of

The text has been modified to "…*Figure 3 (left panel) shows for example the results of fitting the full regression model…*"

P6L3: 'shows as an example the results of' -> 'shows an example of the results of

The text has been modified to "…*Figure 3 (left panel) shows for example the results of fitting the full regression model…*"

---

## Author Comment (AC2) · 7 Dec 2016

We would like to thank the reviewer for the time and the useful comments that helped to clarify important aspects of the model results.

General Comment: This paper examines the Antarctic ozone depletion between 1960 and 1980 in both observations and 17 Chemistry Climate Models (CCMs) based on the REF-B1 scenario from CCMVal2. These models derive an anthropogenic depletion from 1960-1989 between 26.4% and 49.8% of the total period (1960-2000). Observations over the same period suggest a higher depletion of 56%. The paper is clearly written, concise, and adds to the scientific understanding of what the return date choice for "full recovery" implies. I recommend this paper be published after minor revisions (see below).

Specific comments.

Page 3 lines 3-4 All the models certainly do show a 1960-1980 depletion (26.4-49.8%), with approximately six model showing values less than 35%. For these 17 models there was large effort to understand how well these models represented transport, dynamics, and chemistry (i.e., the SPARC Report on the Evaluation of Chemistry-Climate Models, 2010). That is, the range in models is not just due to different dynamical variability. It would be interesting to highlight the models that did better in these process oriented diagnostics in Table 2. This was the approach used in Chapter 2 of the 2010 WMO assessment.

The Discussion and Conclusion section now includes a detailed discussion of the results obtained with respect to the outcome of the SPARC CCMVal evaluation of their photochemistry, transport and UTLS characteristics and the discussion in Chapter 2 of WMO (2010). It turned out that two of the three highest ranked CCMs in SPARC CCMVal (CMAM and WACCM) indeed reproduce the observed total decline between 1960 and 2000 and the relative decline between 1960 and 1980 very well. This has now been highlighted in the text. We included the following paragraph:

"*The observed decrease in total column ozone between 1960 and 2000 was reproduced - within its uncertainty range - by 7 models (CMAM, LMDZrepro, UMSLIMCAT, UMUKCA-METO, UMUKCA-UCAM, WACCM and ULAQ). Two of these CCMs (CMAM, WACCM) obtained the highest ranking in an evaluation of their photochemistry and transport characteristics performed within the SPARC CCMVal activity [SPARC CCMVal, 2010] and discussed in Chapter 2 of the 2010 WMO ozone assessment [WMO, 2011], providing confidence in the robustness of their results. 4 CCMs (AMTRAC3, CNRM-ACM, GEOSCCM, MRI) simulated a stronger ozone decline, and 6 CCMs (CAM3.5, CCSRNIES, EMAC, EMAC-FUB, NIWA-SOCOL, SOCOL) underestimated the observed ozone decline. This divergent model behaviour may be due to the representation of polar ozone chemistry in the models, their dynamical and transport characteristics, or to a combination of both. Based on the detailed evaluation performed as part of the SPARC CCMVal activity [SPARC CCMVal, 2010], we found in our study that the CCMs that represent the observations well, generally (with one exception) show a good potential for chlorine activation and (all) a good representation of chemical ozone depletion in Antarctic spring. CCMs with a stronger ozone loss than observed (cf. Table 2) partly tend to a slight overestimation of chemical ozone depletion (AMTRAC3, GEOSCCM). For some CCMs with weaker ozone decline between 1960 and 2000 a consistent underestimation of chemical ozone depletion was found (CCSRNIES, EMAC, CAM3.5). Thus, the deviations of some CCMs from the observed ozone decline can partly be explained by deficiencies in their polar ozone chemistry. However, in addition, models that underestimate the observed ozone decline were found to suffer from either a too fast transport of air into the Antarctic polar vortex (SOCOL, NIWA-SOCOL) or a too weak insolation of the polar vortex from mid-latitudes in the lower stratosphere (CAM3.5, CCSRNIES, EMAC, SOCOL, NIWA-SOCOL). Both*

*effects lead to lower ESC concentrations by the end of the 20th century in these models (cf. Fig. 1), and as a result an underestimation of the observed polar ozone decline due to ESC. Consistent negative Antarctic ozone changes were diagnosed in the CCMs prior to 1980 as a result of chemical depletion by ESC. This pre-1980 halogen-induced Antarctic ozone depletion amounts to values between 26.4 ± 3.4 and 49.8 ± 6.2 % of the simulated ozone depletion between 1960 and 2000. Hence the CCM simulations are consistent with the observational estimate of a significant EESC induced ozone decline in 1960-1980, albeit nearly all CCMs underestimate the observed decline of 56.4 ± 6.8 %, derived from the NIWA combined total column ozone data base. However, note that the two CCMs, ranked highest in the SPARC CCMVal evaluation of their photochemistry and transport characteristics, CMAM and WACCM, [SPARCCCMVal, 2010] nearly agree with the observed decline between 1960 and 1980 within its uncertainty range.*"

You suggest the the temperature trend in the 1960-1980 period was different in observations relative to most models (Figure 5). Are there any other issues with the models that could explain the lower depletion in this period? E.g., the CCMs used in this study also did not include additional very-short lived bromine (VSL) species. This addition 5-7 pptv of inorganic bromine should contribute to the underestimate the total loss in the 1960-1980 period. It will be interesting if you (not for this paper) redo this analysis for the CCMI models that include this additional VSL bromine source.

Thank you for pointing at this issue. We have added a discussion on the potential effects of VSLS in the Discussion and Conclusion section. The following text was included:

"Another potential reason for the underestimation of the Antarctic ozone decline before 1980 in most CCMs might be the effect on chemical ozone depletion by short-lived bromine compounds of natural biogenic origin, so-called very short lived substances (VSLS). The effects of VSLS which contribute 20-30 % to the present-day stratospheric bromine content [WMO, 2011] on Antarctic stratospheric ozone were not included in the CCMVal-2 simulations. Braesicke et al. [2013] and Sinnhuber and Meul [2014] showed that taking brominated VSLS in their CCMs into account leads to a significant reduction of Antarctic polar ozone. In a transient REF-B1 simulation using the same FUB-EMAC CCM as included in this study but with prescribed VSLS sources, Sinnhuber and Meul [2014] found a reduction of October mean ozone in the lower Antarctic stratosphere of more than 20 %. However, although constant VSLS emissions were prescribed over the whole simulation period of 1960-2005, the impact of VSLS was stronger in the most recent period after 1980 with enhanced chlorine due to combined bromine-chlorine catalytic ozone loss cycles. Hence, including the VSLS effect leads to an enhancement of the 1960-2000 Antarctic ozone depletion, but reduces the relative change in 1960-1980 compared to the full period. Further insight is expected from the analysis of the new CCMI- simulations that will include the effects of VSLS."

Page 3, equation 1. The authors did a very nice job of explaining the approach of determining the degree of halogen-induced ozone for the 1960-1980 period. Question: the temperature anomaly at 100hPa is used in the regression fit to address dynamical variability. What equation (1) does not address is the sensitivity of the ozone chemistry in the model to absolute biases in temperature. It therefore would be very informative to show the lower polar stratosphere absolution temperature evolution similar to Figure 4. E.g., if two models both show a similar representation of ESC (i.e., consistent transport/dynamics) and a similar absolute temperature trend (new figure), and would happen to have a different temperature trend vs ozone trend sensitivity (Figure 5) – I believe this would highlight issues in the chemistry representation between the two models. Generally, it would be nice to comment on how this technique could be used to evaluate model components.

A new Figure 6 has been added showing the evolution of absolute temperature at 100 hPa where chemical ozone depletion is strongest. The individual CCM temperature curves have not been

adjusted to a common basis in 1960, revealing the large spread in absolute temperature between the models. In the Discussion and Conclusion section an additional paragraph discussing the role of temperature for ozone transport and chemistry has been added on P6, L29::

*"Figure 5 shows the evolution of the polar cap mean SON mean temperature at 100 hPa between 1960 and 2000, fitted with piecewise linear trends from 1960 to 1980 and from 1980 to 2000. While the Antarctic lower stratosphere temperature observations showed warming in SON from 1960 to 1980, the CCMs span a wide range of trends, indicating different temperature trend regimes resulting from model dynamical variability differing from what happened in reality. In addition, the absolute temperature values differ between the CCMs by more than 10 K, which directly affects the potential for chemical ozone loss in the models."*

Page 4. The discussion of how the observations are combined are in reasonably detailed. Based on this discussion and use of equation 1 (and 2) this work suggests a decline of 56.4 +-6.8%. Maybe I missed it, but how did you come up with an uncertainty value for this decline (i.e., +-6.8%)?

For both the model data and the observations, the uncertainty values for the regression model fit coefficient were calculated using standard error propagation. The following sentence has been added to the text at the end of Section 2:

*"Uncertainty values of the total column ozone and further derived quantities have been calculated by applying the standard formulae for error propagation to determine the uncertainties on the regression model fit coefficients."*

Also, since the results are for SON mean total ozone polar cap average (e.g., Figure 2) – are you masking the model results for periods that are in the dark with little ozone depletion (e.g., high latitudes in September)? That is, are you treating the model and observations derivation of the 1960-1980 decline in a consistent manner?

We agree that the underestimation of the ozone loss in the models might be related to a sampling bias of the CCMs that include high latitudes with little ozone depletion in their polar cap mean, while the observations are made in sunlight. However, this is only an issue in September. Moreover, the region of the polar cap that is in perpetual darkness is very small (as a fraction of the whole area poleward of 60°S) and shrinks to zero by the end of September. So, this effect should be of the order of a few percent. We included the following sentence in the article:

*"Apart from the chemical and dynamical performance of the models, their underestimation of the ozone loss might be related to a sampling bias of the CCMs that include high latitudes with little ozone depletion in their polar cap mean, while the observations are made in sunlight. However, this bias exists in September only. Moreover, the region of the polar cap that is in perpetual darkness is very small (as a fraction of the whole area poleward of 60°S) and shrinks to zero by the end of September. So, this effect should be of the order of a few percent."*

Figure 2 caption is missing the prime symbol in "c T".

The prime has been added in Figs 2 and 3.